# Colour change and behavioural choice facilitate chameleon prawn camouflage against different seaweed backgrounds

Samuel D. Green [1,4], Rafael C. Duarte [2,3,4], Emily Kellett[1], Natasha Alagaratnam[1] & Martin Stevens [1]

Camouflage is driven by matching the visual environment, yet natural habitats are rarely uniform and comprise many backgrounds. Therefore, species often exhibit adaptive traits to maintain crypsis, including colour change and behavioural choice of substrates. However, previous work largely considered these solutions in isolation, whereas many species may use a combination of behaviour and appearance to facilitate concealment. Here we show that green and red chameleon prawns (*Hippolyte varians*) closely resemble their associated seaweed substrates to the vision of predatory fish, and that they can change colour to effectively match new backgrounds. Prawns also select colour-matching substrates when offered a choice. However, colour change occurs over weeks, consistent with seasonal changes in algal cover, whereas behavioural choice of matching substrates occurs in the short-term, facilitating matches within heterogeneous environments. We demonstrate how colour change and behaviour combine to facilitate camouflage against different substrates in environments varying spatially and temporally.

[1] Centre for Ecology and Conservation, University of Exeter (Penryn Campus), Cornwall TR10 9FE, UK. [2] Centro de Biologia Marinha, Universidade de São Paulo, São Sebastião 11612-109, Brazil. [3] Centro de Ciências Naturais e Humanas, Universidade Federal do ABC (UFABC), São Bernardo do Campo 09606-045, Brazil. [4] These authors contributed equally: Samuel D. Green, Rafael C. Duarte. Correspondence and requests for materials should be addressed to S.D.G. (email: s.green4@exeter.ac.uk)

Cryptic coloration allowing visual camouflage is a cosmopolitan antipredator strategy in nature and provides classic examples of evolution by natural selection[1–5]. Crypsis works by reducing the chance of prey detection or recognition by the visual system of potential predators[1,6,7], often through background matching[1,5,8–10]. However backgrounds within natural habitats are rarely uniform and vary considerably in colour and pattern across space and time[11]. Therefore, an array of adaptations exist to overcome this problem, including generalist coloration, colour polymorphisms, ontogenetic changes, colour change and behaviourally oriented choices[1,3,12–14]. Of these strategies, colour change is widespread in nature and allows individuals of many species to adjust the colour aspect of their camouflage to both the environment and to the visual systems of different predators[13,15]. Research has tended to focus on animals capable of rapid change (i.e. seconds/minutes) as opposed to slower changes (days/weeks), despite the latter likely being more prevalent across taxonomic groups[13,15]. The benefit of rapid colour change for crypsis in many animal groups is clear (e.g. chameleons[16], cuttlefish[17] and fish[18–21]) since it provides a response to environments that change unpredictably in the short term[11]. The function of slower changes can be less intuitive[13], although alterations occurring over a period of days–weeks likely improve concealment in response to predictable and slow environmental changes or associated with life history[11]. This may include seasonal changes in habitat availability. The drivers and outcome of longer-term colour change for camouflage to predator vision has rarely been properly quantified.

Aside from direct colour adaptations, animals may also facilitate camouflage by behaviourally choosing appropriate backgrounds[22–28]. Despite the intuitiveness of this idea there has historically been a lack of consistent experimental investigation into this area[27], and few rigorous tests of how and when behavioural choices facilitate crypsis on natural substrates have been conducted. However, recent work has demonstrated the existence of background choice at an individual level for improving concealment. Both the nesting substrate preference of laboratory Japanese quail (*Coturnix japonica*)[26] and the resting spots of wild Aegean wall lizards (*Podarcis erhardii*)[22], for example, have been shown to be closely tied to individual appearance of eggs and adults, respectively. In addition, ground-nesting birds from nine different species improved their level of camouflage by choosing appropriate backgrounds across multiple spatial scales[29]. Recent work also demonstrates how moths change their resting posture based on individual levels of camouflage[28].

However, most studies to date have tested background choices in species where individuals are fixed in appearance, and work has yet to fully consider the importance of behavioural preferences affecting concealment of species capable of colour change (but see refs. [18,30,31]). From a conceptual perspective, individuals from species capable of rapid colour alterations might be under less intense selective pressure to rely on behavioural choices for crypsis, since they can rapidly alter appearance regardless of the background[32]. For example, cuttlefish (*Sepia officinalis*) do not exhibit substrate preferences for camouflage, but rely on visual environmental cues to adopt camouflage patterns[33]. Conversely, individuals from species capable of changing colour over a longer period of time may depend heavily on an ability to select visually appropriate backgrounds. In this manner, individuals may maintain a level of protection by reducing the level of mismatch during the process of colour change[11,27]. However, the relationship between these two strategies is more complex and likely depends on a number of additional factors. The larvae of two different salamander species, for example, change colour over a few hours when placed in mismatched

substrates, but the capacity for choosing concealing backgrounds differed between species and was affected by predation risk[34]. Also, rock gobies (*Gobius paganellus*) are able to rapidly alter coloration to improve camouflage, but individuals also selected darker backgrounds that improved matching, indicating that a combination of behaviour and colour change may reduce limitations of each strategy[18]. Similarly, behavioural choices for colour-matching backgrounds potentially improving camouflage were also demonstrated in fish species, both in the laboratory[31] and field[19]. Therefore, colour change and oriented-choice behaviours are non-mutually exclusive traits that are widely utilised by cryptic species to maximise colour concealment and reduce prey detection by predators[18,27,30].

Crustaceans, particularly those living in dynamic intertidal habitats, have frequently been used as key organisms to investigate the fitness advantages conveyed by cryptic traits[35,36]. Within this group, caridean shrimps are well known to exhibit remarkable variation in coloration often allied to a cryptic lifestyle[24,37–43]. The chameleon prawn (*Hippolyte varians*) is a small species that exhibits a variety of colour forms that, subjectively, show a close similarity to the seaweed species on which they associate (Fig. 1). Research at the end of the nineteenth century revealed that this species was capable of changing colour over different time-scales, including driven by light intensity, periodic shifts between diurnal/nocturnal forms and slow changes in response to novel

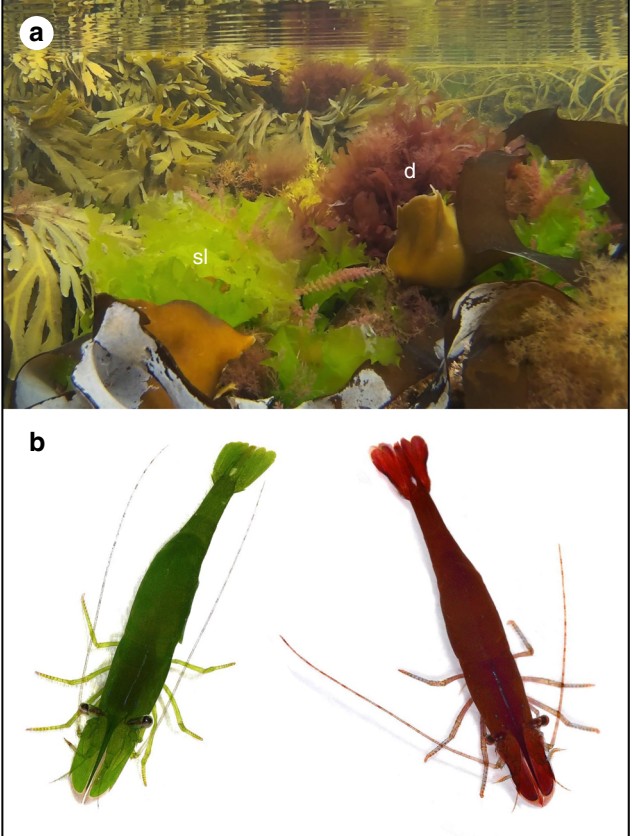

**Fig. 1** *Hippolyte varians* seaweed habitats and colour types. Chameleon prawns (*H. varians*) are found in a variety of colour forms which, subjectively, show a close resemblance to the seaweed species on which they are found. **a** The heterogeneous nature of intertidal rock pool habitats around UK shores, including the green algae sea lettuce (*Ulva lactuca*—sl) and the red algae dulse (*Palmaria palmata*—d). **b** The green and red colour types of chameleon prawns used in this study. Prawn images provided by Dr. Cyril Bennett

substrates[39,40,43]. In addition, the different colour forms of chameleon prawns may choose concealing algal substrates when displaced[44]. However, there is little information on how colour change and behavioural traits may operate in tandem in order to improve camouflage and reduce detection by predators over different spatial and temporal scales.

Here, we use digital photography, image analysis and vision models of ecologically relevant predators (fish) to quantify how chameleon prawn camouflage is mediated by colour change and behavioural choices. We focus on the homogenous green and red prawns (Fig. 1) against natural seaweed substrates: the green sea lettuce (Ulva lactuca) and the red dulse (Palmaria palmata). We first assessed the level of camouflage achieved by the different prawn forms against seaweed in the field, followed by laboratory and field experiments to test whether prawn camouflage is improved by longer-duration colour change, and whether prawns show behavioural preferences for matching seaweed types, respectively. We expected that if coloration is adaptive and increases prawn survival, the different colour forms will exhibit increased camouflage against their main seaweed substrate. Similarly, we expected that colour change will induce alteration of prawn coloration towards the colour of the new substrate, reducing the level of colour mismatch; and that prawns will show a strong behavioural preference for matching seaweed backgrounds when given a choice. Our results clearly demonstrate that, to predator vision, prawns exhibit effective camouflage against their associated seaweed substrate and are able to alter coloration improving camouflage in response to changes to their background. Additionally, prawns show strong behavioural preferences for selecting substrates that match their own appearance. Our findings support the notion that, since longer-duration colour changes inevitably lead to some degree of mismatch between individuals and backgrounds, slow-change species likely have evolved convergent behavioural strategies to ensure that a cryptic benefit is maintained in a species overall cryptic stratagem.

## Results

**Coloration provides camouflage against natural substrates.** We modelled the coloration of both green and red prawns and natural seaweed substrates (green sea lettuce and red dulse) to predator vision using the spectral sensitivity data of the dichromatic pollack (Pollachius pollachius)[45] and the trichromatic two-spotted goby (Gobiusculus flavescens)[46]. We used a polynomial mapping function to convert prawn and seaweed multispectral images from the camera colour space to fish vision[47,48], generating values for pollack and goby predicted cone catches (see Methods).

In order to assess the level of camouflage between prawn colour types and seaweed species from the modelled perspectives of fish vision we used the widely implemented noise model for colour discrimination of Vorobyev and Osorio[49], which predicts chromatic signal discriminability as units of just noticeable differences (JNDs). Colour JND comparisons across combinations of colour type and seaweed species showed type-specific colour matches to seaweed species for both pollack and goby on distinct backgrounds (LMM: pollack: $F_{1,252} = 544.18$, $p < 0.001$; goby: $F_{1,252} = 339.27$, $p < 0.001$). Specifically, colour contrasts of green prawns against green sea lettuce (mean ± SE = 1.73 ± 0.15 for pollack, and 1.91 ± 0.17 for goby vision) were around the threshold of discriminability and much lower than against red dulse (6.66 ± 0.34 for pollack, and 6.38 ± 0.36 for goby vision). Concurrently, the opposite pattern was found for red prawns, with individuals showing small colour differences against red dulse (mean ± SE = 1.44 ± 0.12 for pollack, and 2.48 ± 0.19

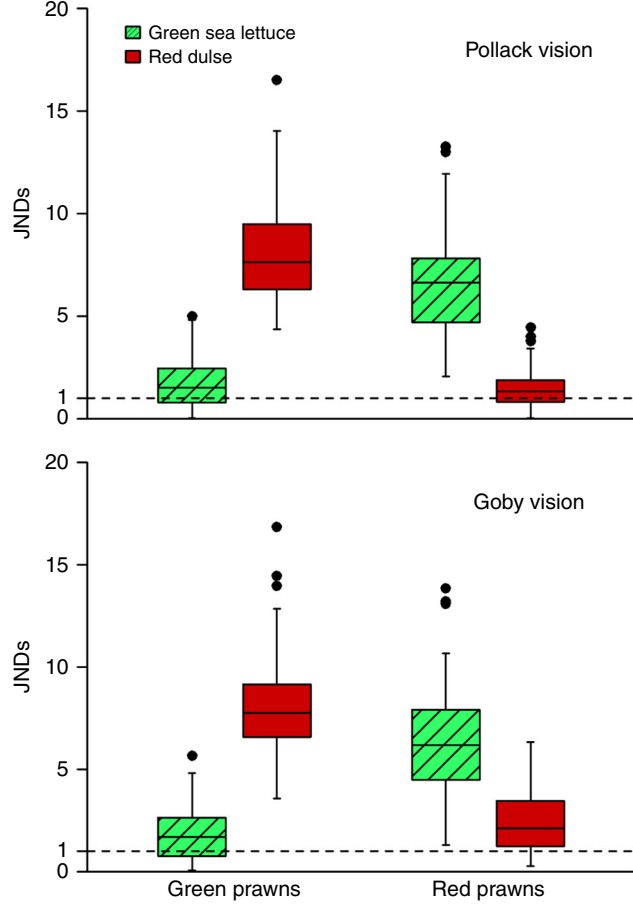

**Fig. 2** Coloration provides adaptive camouflage against natural substrates. Pollack (upper panel) and goby (lower panel) visual discrimination (as just noticeable differences; JNDs) of green ($n = 64$) and red ($n = 64$) prawn types against green sea lettuce (Ulva lactuca) and red dulse (Palmaria palmata) habitats. Boxes display medians and inter-quartile ranges (IQRs), whiskers represent lowest and highest values within 1.5 × IQRs, and black filled circles represent outliers. The dashed line (JND = 1) indicates the threshold for predicted visual discrimination of prawns by fish predators

for goby vision) and higher contrasts against green sea lettuce (8.04 ± 0.31 for pollack, and 8.15 ± 0.33 for goby vision) (Fig. 2). This shows that green and red prawns are a very close match to their respective substrates in the field.

**Prawns change colour over time in response to new substrates.** In order to test the ability of H. varians to change colour in response to new substrates, we conducted a laboratory experiment where both colour forms were kept on colour-contrasting seaweeds over a period of 30 days. Hue values (i.e. a measure of colour type, see Methods for details) of green prawns kept on red dulse significantly increased over time to both pollack (LMM: $F_{6,124} = 51.03$, $p < 0.001$) and goby vision (LMM: $F_{6,124} = 45.65$, $p < 0.001$). Conversely, hue of red prawns maintained on green sea lettuce significantly decreased over time to both visual systems (pollack: $F_{6,118} = 65.48$, $p < 0.001$; goby: $F_{6,118} = 44.31$, $p < 0.001$—Fig. 3). These results can be explained by changes in the relative proportion of reflectance in the short-wave channel of prawns over time. At the beginning of the experiment, green prawns have relatively low short-wave reflectance compared to red prawns. However, after a few days exposed to colour-contrasting seaweed, there was a crossover in this pattern, with

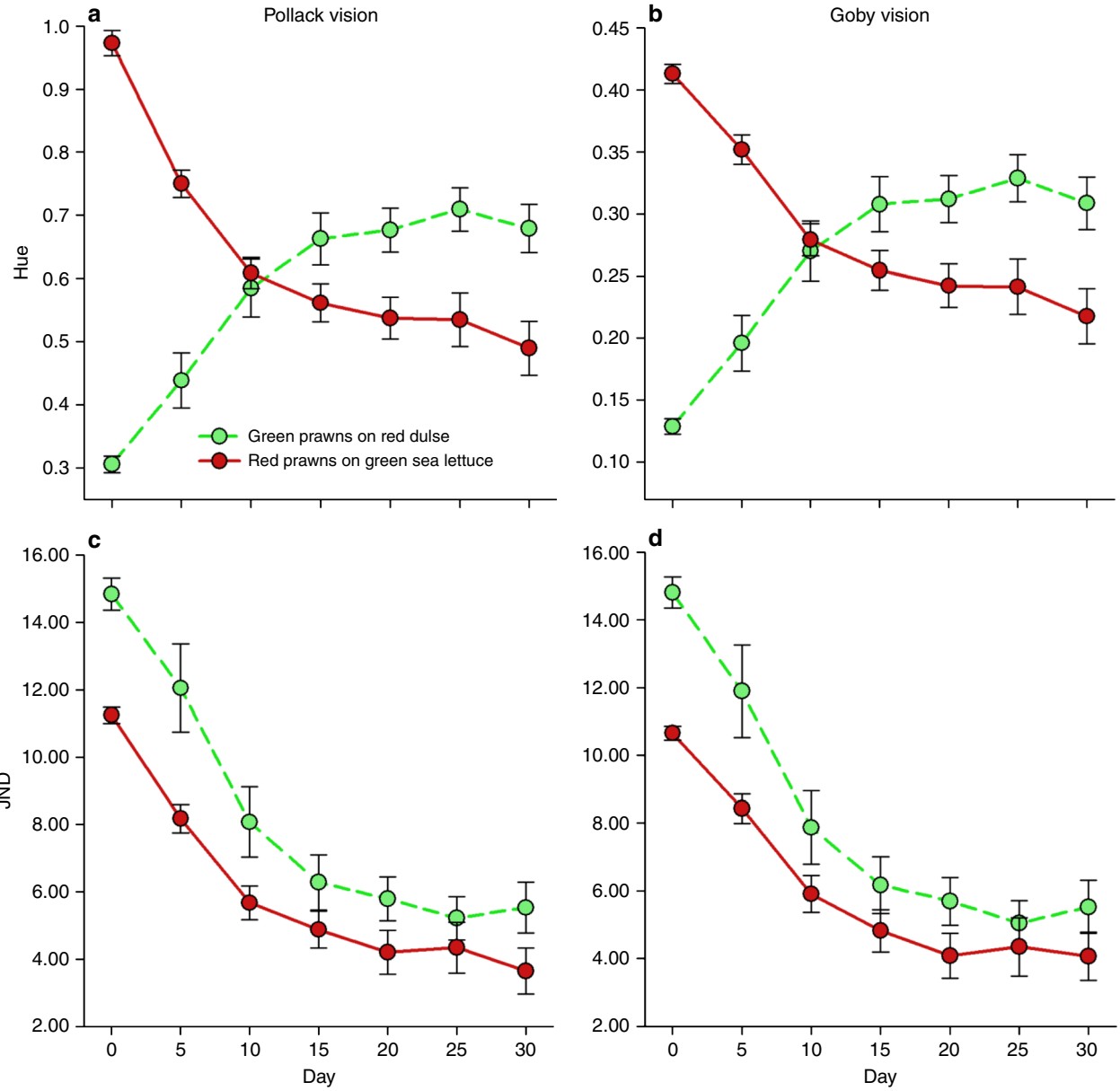

**Fig. 3** Colour change improves camouflage match in response to mismatched substrates. Changes in hue (mean ± SE) of green ($n = 25$) and red ($n = 25$) prawns when placed against seaweed of opposing coloration over 30 days to the vision of both pollack (**a**) and goby (**b**). A crossover pattern can be observed between the two colour types as hue in green prawns increases due to an increase in the relative proportion of short-wave reflectance, whereas hue for red prawns decreases due to the reduction of reflectance in this channel. Changes in JNDs (mean ± SE) over time for green and red prawns to the modelled perspectives of (**c**) pollack and (**d**) goby, demonstrating that both colour types are able to alter their coloration in response to new substrates and that this change increases the level of camouflage (lower JNDs) when perceived by ecologically relevant predators

green prawns exhibiting larger short-wave reflectance than red individuals (Fig. 3). These changes resulted in green prawns kept on red dulse becoming redder, and red prawns kept on green sea lettuce becoming greener, to human vision (Fig. 4). Hue changes of both colour types occurred faster in the initial 10 days of the experiment (comparisons between days 0, 5 and 10 all significant—Tukey's *t* tests; $p < 0.05$), becoming smaller and nonsignificant over the next 20 days (Fig. 3). The body size of individuals, included as a covariate in the model, was significantly correlated with hue only for green prawns (pollack: $F_{1,22} = 16.21$, $p < 0.001$; goby: $F_{1,22} = 16.18$, $p < 0.001$), indicating that larger green individuals change less for hue along time compared to smaller prawns.

**Colour change enables better camouflage on new substrates.** Overall, the colour JNDs of green and red prawns against red dulse and green sea lettuce significantly reduced over time to both pollack (LMM—green prawns: $F_{6,124} = 44.16$, $p < 0.001$, day 0 [mean ± SE] = $14.83 ± 0.48 \rightarrow$ day 30 = $5.53 ± 0.76$; red prawns: $F_{6,118} = 49.44$, $p < 0.001$, day 0 = $11.25 ± 0.25 \rightarrow$ day 30 = $3.65 ± 0.68$; Fig. 3) and goby vision (LMM—green prawns: $F_{6,124} = 44.68$, $p < 0.001$, day 0 = $14.81 ± 0.46 \rightarrow$ day 30 = $5.52 ± 0.79$; red prawns: $F_{6,124} = 47.33$, $p < 0.001$, day 0 = $10.66 ± 0.20 \rightarrow$ day 30 = $4.07 ± 0.71$; Fig. 3). These results show that both prawn types increased their chromatic similarity to the new seaweed background over time, improving their camouflage (Fig. 4). Similar to hue, prawn size was significantly correlated

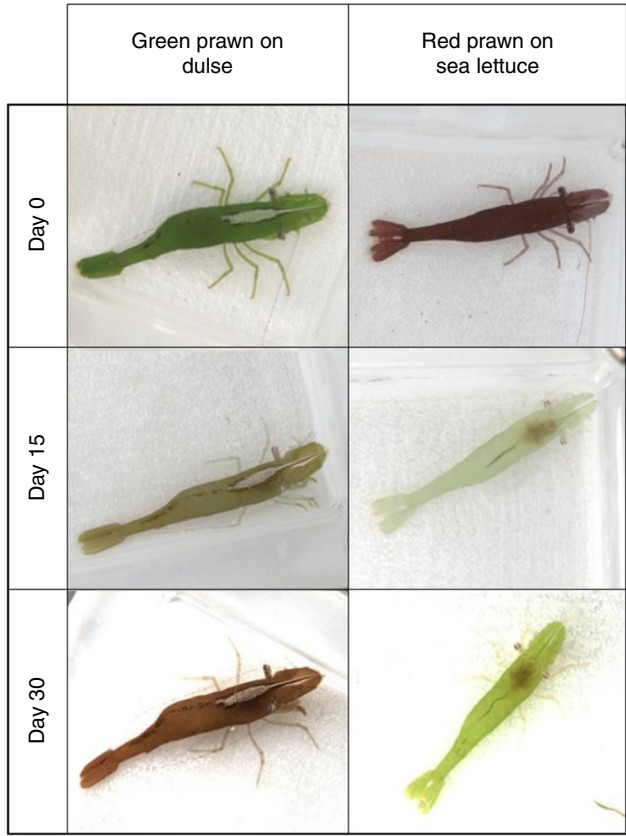

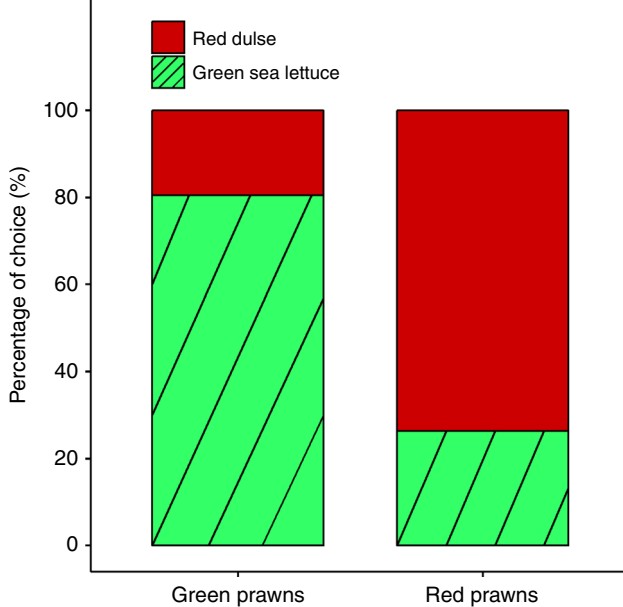

**Fig. 5** Behavioural substrate choices facilitate crypsis. Percentages of choices ($n = 79$) made by green and red chameleon prawns (*Hippolyte varians*) between two seaweed species, green sea lettuce (*Ulva lactuca*) and red dulse (*Palmaria palmata*). When given a choice, each colour type clearly chooses the substrate of corresponding coloration (e.g. green—green sea lettuce; red—red dulse)

**Fig. 4** Colour change in chameleon prawns. Changes in the body coloration of green and red chameleon prawns (*Hippolyte varians*) in response to colour mismatching seaweed backgrounds over the 30 day experiment. For the purposes of display here, images were linearised and equalised but not converted to predator vision

with JNDs only for green prawns (LMM—pollack: $F_{1,22} = 16.42$, $p < 0.001$; goby: $F_{1,22} = 15.45$, $p < 0.001$), with larger individuals being more poorly concealed against red dulse (i.e. exhibiting larger JNDs) than small prawns over the experiment.

**Behavioural substrate choice facilitates crypsis.** We performed a series of behavioural trials using a Y-choice decision chamber (see Methods) to determine whether prawns actively selected a background that improved their level of camouflage. For each trial, lasting 10 min, we gave a choice between red dulse and green sea lettuce for both green and red prawns. Green prawns were significantly more likely to choose green sea lettuce than red dulse (Exact binomial test: proportion = 0.80, $n = 41$, $p < 0.001$), while red prawns preferred red dulse instead of green sea lettuce (Exact binomial test: proportion = 0.74, $n = 38$, $p = 0.005$). Therefore, prawns show strong behavioural preferences for seaweed matching their current coloration (Fig. 5).

### Discussion

Our results show that the coloration of chameleon prawns provides effective visual camouflage to predator vision against their main seaweed substrates. We first quantified the level of in situ camouflage between prawns and seaweed using vision models of two fish predators, showing that prawn concealment was closer and more effective against the algal substrate against which they would reside. Next, we show that green and red prawns change colour over time when placed on mismatched seaweed and improve their camouflage on the new substrate. Prawn coloration

is therefore highly plastic, with prawns able to switch from red to green and vice versa over a period of weeks. Finally, we show that individuals actively choose a background based on their current coloration that improves their level of camouflage to predator vision.

The capacity to change appearance across species has likely evolved to cope with both spatial and temporal uncertainty over a short time frame, or with predictable changes over a longer time period[11]. In both cases, it enables animals to change their appearance as they move between patches within their environment (e.g. spatial heterogeneity) or as the composition of environment changes around them over time (e.g. temporal heterogeneity)[11]. In the case of chameleon prawns, our results indicate that colour change is unlikely to have evolved as response to the spatial heterogeneity of their habitat, as is the case for rapidly changing animals such as chameleons[16] and cuttlefish[32]. Instead, the slower colour change of chameleon prawns likely allows individuals to maintain their camouflage in response to seasonal variation in the abundance of seaweed species, in accordance with more predictable patterns of environmental variation[11,13]. In contrast to slow colour change, our results from the behavioural experiment show that the ability of chameleon prawns to select appropriate backgrounds is likely a key strategy for maintaining camouflage in the short term and to cope with the considerable spatial variation in the habitat where individuals live[27]. Oriented choices will also be important to help prawns dealing with some unique challenges of the intertidal environment, such as wave action dislodging individuals from preferred substrates and tidal changes influencing habitat availability over the day.

Our results also indicate that the effectiveness of colour change for camouflage was higher for small green prawns compared to larger individuals. This relationship needs to be properly investigated in future studies, but speculatively could indicate that larger green prawns have less selection acting on them due to

more effective escape behaviours or by achieving a size refuge from predators, or due to physiological limitations. Why this occurs only for green prawns is difficult to explain but may be related to the fact that red prawns when changing to green always exhibit lower JNDs compared to the opposite (Fig. 3). This seems to be a physiological constraint, since the red coloration is probably defined by the presence of red-yellow pigments within chromatophore cells, while the green tone is provided by the presence of only the yellow pigment (similar to that observed in the prawns *Heptacarpus pictus* and *H. paludicola*[37]). Therefore, changing from red to green may be easier and faster than the opposite since both pigments (i.e. red and yellow) are already present within the colour cells of red prawns. On the other hand, green prawns changing to red would need to metabolise red pigments (probably by food ingestion[41]), which would take more time, especially for larger individuals, potentially explaining the higher JNDs during the colour change process and the size effects we observed.

Seasonal variation in animal appearance in response to changes in substrate availability is frequently observed in nature[11]. Many birds and mammals, for example, change their coat colour from brown in the summer to completely white in the winter as response to the appearance of snow[11]. In addition, populations of the polymorphic pacific tree frog (*Hyla regilla*) are characterized by both fixed and colour changing morphs and the maintenance of such colour variation in the population is associated with changes in microhabitat use of individuals due to seasonal changes in substrate availability[50]. The assemblage of seaweed species within the intertidal zone varies through the year as a function of both the species' life history and some environmental conditions. The red dulse (*Palmaria palmata*) is a perennial species and, while it undergoes a seasonal burst in growth over the summer months, its holdfasts and fronds provide a 'fixed' habitat over a period of several years[51]. On the other hand, the green sea lettuce (*Ulva lactuca*) has a pseudo-perennial life cycle in which the basal portion but not the fronds survive over the years. In this case, seaweed biomass and therefore the habitat availability for algal-dwelling species fluctuates over the year[51,52]. Sea lettuce exhibits a marked period of rapid growth during the warmer months and, although it may be found throughout the year, it is more susceptible to the effects of harsher winter weather (e.g. lower temperatures, storms and currents) in shallow regions such as rock pools[52]. As such, the combination of slower colour changes and behavioural habitat preferences may enable chameleon prawns to maintain the benefits of cryptic coloration, meanwhile allowing the species to take advantages of seasonal abundances of algal habitats throughout the year. Camouflage through colour change in the carnival prawn (*Hippolyte obliquimanus*), which associates with different seaweeds along the Brazilian coast, is also related to seasonal fluctuations in the cover of its main habitat, the brown algae (*Sargassum furcatum*)[41]. During the summer, this seaweed dominates the shallow rocky areas in southeast Brazil and brown prawns attain the largest densities[53]. However, in winter, *Sargassum* cover decreases and the density of the different colour types in the population changes considerably, following the fluctuation of seaweed habitats (Duarte and Flores, unpublished data). In addition to seasonal changes in substrate, intertidal species may undergo seasonal shifts in predation pressures as fish species move inshore and as juveniles develop, and future work could quantify how the level of crypsis may vary with these predator shifts.

Phenotype–environment matching is assumed to be a common outcome of selection for cryptic traits, yet most research to date has shown indirect associations between animal phenotypes and habitats (i.e. has not quantified camouflage itself, but see ref. [35]).

Our work demonstrates that camouflage in chameleon prawns is enhanced on substrates where they live, and that this close association between phenotype and habitat is predicted to be effective to predator vision. Studies investigating associations between the appearance of juvenile shore crabs (*Carcinus maenas*) and that of their habitat substrate composition over a range of spatial scales have demonstrated the strongest associations at the micro-scale (<1 m²)[54]. While camouflage is dependent on the appearance match between individuals and their local habitat, an animal may improve this by either reorienting its body relative to its background or by selecting a more appropriate substrate[27]. Indeed, many individuals from the same or different species have evolved preferences for habitat patches that enable increased levels of camouflage[27]. For highly mobile species, it is likely that these behavioural preferences allow for the active maintenance of phenotype–environment associations within heterogeneous habitats. For species capable of colour change, we would expect that behavioural preferences for substrates would change in tandem with a change in body coloration to maintain the selective advantages conveyed by visual camouflage[27]. In addition, we might also expect other processes to come into play for maintaining colour variation, including multiple morph types acting to hinder predator search image formation, and frequency-dependent selection[55].

Changes in behavioural preferences mediated by modifications of body coloration have also been demonstrated in guppies (*Poecilia reticulata*), in which individuals spent significantly more time in black and white habitat zones after being induced to change colour in corresponding black and white tanks[56]. Future work should further consider coloration and camouflage with regards to predator vision and measured attack rates. Another research avenue is to understand how predator cues may affect colour change and cryptic behaviours. For example, in the presence of a perceived predation threat, animals may improve their capacity to change colour and select concealing backgrounds, the latter approach especially in slow colour change species. In salamander larvae, the addition of predator cues in experimental tanks increases larval preference for dark backgrounds followed by a corresponding change in individual coloration[34]. However, in the absence of predator cues, larvae spend equal time in light and dark habitat zones, adopting a more intermediate colour form[34]. In the case of chameleon prawns, we would expect that the addition of predator cues may speed up the colour change process and lead to an increase in the proportion of prawns making a choice for concealing substrates.

In the case of polyphenic species, intra-specific variation in coloration and behaviour may allow different individuals to utilise distinct aspects of visual camouflage to adopt alternative life-histories[57]. Duarte et al[41] demonstrated differences between colour morphs of the 'carnival' prawn (*Hippolyte obliquimanus*) in algal preference, showing morph-specific differences in morphology and mobility, which indicate contrasting benthic/pelagic lifestyles[41]. Homogenous coloured morphs showed greater habitat fidelity and a stouter body shape, whereas transparent morphs displayed a more streamlined body shape and increased levels of swimming activity[41]. In the case of chameleon prawns, besides the homogenous coloured forms we studied here (Fig. 1), there exists an assortment of alternative forms that combine colour patterns (spots or stripes) with some degree of transparency[13,43], and these may also reflect different camouflage, behaviour and life history strategies. In our study, the visual models used are based on colour perception and the spectral sensitivities of ecologically relevant predators available in the literature[45,46]. However, we do not model the spatial acuity of the predators, which is relevant to pattern matching and something that may be especially relevant to transparent prawn types with their intricate markings.

Although our results clearly show that prawns choose backgrounds that improve camouflage, there are many questions regarding what cues control preferences for certain substrates. There is limited information about the existence of colour vision in similar crustaceans, which limits our understanding of whether chameleon prawns are able to identify different seaweeds based on colour cues. Alternatively, caridean prawns living on the pelagic seaweed Sargassum natans select appropriate backgrounds based on their shape, with individuals preferring habitats containing structures that best matched their body shape[24]. Therefore, in some cases, the structure of the habitat allied with a range of behavioural adaptations in the use of that structure may provide better protection from predators than concealing coloration[58]. Since our knowledge of chameleon prawn visual capabilities is limited, it may be that individuals depend upon identifying the structural form of their preferred habitat when making a choice either independently of, or in conjunction with, its coloration. Finally, there is growing evidence about the importance of chemical cues and habitat complexity regulating habitat choices in a wide array of marine organisms, especially for those living on biological substrates (e.g. seaweeds, corals)[59]. Future work could aim to quantify the importance of visual components and other sensory cues that species use when identifying suitable substrates for crypsis.

A wide range of mechanisms and evolutionary pressures control appearance and the adaptive benefits of colour forms in polymorphic/polyphenic populations, and the fine-tuning of a species' cryptic stratagem may depend on the integration of morphology, behaviour and the environment itself[13,29]. Here, we show that chameleon prawns are able to alter body coloration to improve camouflage against new substrates, potentially allowing prawns to exploit seasonal changes in resource abundance (e.g. food and shelter[43]). This would allow the exploitation of a wider range of resources within structurally complex habitats, potentially reducing intra-specific competition[60] and predation risk[61]. Concurrently, behavioural preferences facilitate camouflage over time-scales when colour change is too slow. The growing number of studies testing how combinations of chromatic (particularly colour change) and behavioural traits influence crypsis, and the fact that the above-mentioned traits are displayed by a range of phylogenetically and ecologically distinct systems, is indicative of the convergent evolution of these cryptic strategies and the importance of adaptive benefits conveyed to species in order to maintain crypsis in heterogeneous habitats in wild systems.

## Methods

**Prawn and seaweed sampling.** Chameleon prawns (Hippolyte varians) of varied sex and size, and two seaweed species, green sea lettuce (Ulva lactuca) and red dulse (Palmaria palmata), were sampled from rocky tide pools during low-tide periods in the intertidal zone of Gyllyngvase beach, Falmouth, Cornwall, UK (50°08′33″N, 05°04′08″W). Prawns and algae for the initial camouflage analysis were collected during the spring-summer of 2015, 2016 and 2017, while those for the behavioural trials and colour change experiment were collected in the autumn of 2016. Prawns were obtained by vigorously shaking seaweeds in grey buckets or by dunking buckets adjacent to seaweed patches, after which they were visually classified in green or red colour types (Fig. 1). After sampling, those prawns used in colour change experiments were individually housed inside (2.5″× 2.5″) white guttering containers with ultra-fine insect mesh netting bases suspended within two indoor tanks with their original seaweed hosts (i.e. green prawns on green sea lettuce and red prawns on red dulse). The water in the tanks was kept at a salinity between 30 and 35 ppm using Aquarium Systems Instant Ocean sea salt mixture (Swell Ltd, Cheshire, UK) and maintained at a constant temperature (matching local sea temperature ~15 °C) with a Swell D&D DC300 Refrigerated Cooler (Swell Ltd, Cheshire, UK). The water was pumped and filtered using an Eheim Classic 350 2215 External Filter (EHEIM GmbH & Co. KG, Germany) and 13 mm aquarium tubing. Each guttering container was fed with oxygenated water from a PVC piping system above the tank. Natural lighting was simulated using two GroBeam 600 Ultima ND Natural Daylight and one AquaBeam 600 Ultima NUV (Tropical Marine Centre, Hertfordshire, UK). These lights were set to a 12-light cycle from

07:30 to 19:30. Different sets of prawns and seaweed were used for the three different procedures we carried out, which included calculating camouflage of wild prawns against seaweed to test whether prawns are better concealed to the backgrounds they are found, and performing colour change experiments and behavioural choice trials to test whether prawns are capable to change colour and/or choose backgrounds to improve their concealment and camouflage. The work was approved by University of Exeter Bioscience ethics committee (code: 2017/1568).

**Photography.** We used digital image analysis to obtain colour estimates of prawns and seaweed in all the different experimental procedures we carried out. Pieces of both green sea lettuce and red dulse, as well as living chameleon prawns of both green and red colour types, were photographed in a dark room in a custom-made acrylic chamber ($5 \times 5$ cm$^2$) using cameras converted to full-spectrum sensitivity by removal of the ultraviolet (UV) and infrared (IR) blocking filter (Advanced Camera Services Limited, Norfolk, UK). Initial camouflage images were taken using a Nikon D7000 digital camera fitted with a Coastal Optic 60 mm lens, and images for the colour change experiment were taken using a Nikon D90 SLR camera fitted with a 105 mm Nikkor lens. Human-visible spectrum photographs were obtained through a Baader UV–IR blocking filter (Baader Planetarium, Mammendorf, Germany), permitting only visible spectrum light (420–680 nm), and UV photographs were taken with a Baader UV pass filter allowing UV light (320–380 nm). Following initial imaging, UV photographs were not taken of prawns during the colour change experiment to speed up proceedings, since both predator visual systems do not have UV-sensitive vision[45,46] and prawn/seaweed general coloration is low in UV reflectance. Illumination was provided by one human-visible Arc Lamp (70 W, 6500K Iwasaki Colour Arc Lamp) with a modified bulb in order to remove its UV filter enabling UV photography. We also placed a PTFE (polytetrafluoroethylene) diffuser cylinder around the photography chamber in order to ensure even lightning conditions and reduce light reflection. All images were taken in RAW format, with manual white balancing and fixed aperture settings, to avoid overexposed areas[47] and included a black and a white Spectralon reflectance standards (Labsphere—8.5 and 95% for the initial camouflage data, and 7 and 93% for the colour change data) and a scale bar. After capture, images were linearised using curves modelled from eight Spectralon reflectance standards with reflectance values ranging from 2 to 99% in order to correct for camera non-linear responses to light intensity[47,48]. Each photograph was also equalised for any changes in light conditions using the two Spectralon standards and saved as 32-bit multispectral images. All these routines were performed by a series of customised functions implemented in the ImageJ software[48].

**Visual modelling.** In our study, we choose fish as our model predators as they are a common group associated with intertidal areas either in association with seaweed and seagrasses or as opportunistic visitors coming from deeper areas during high tide periods. Since the assemblage of fish species in a certain area will vary depending on different factors, including local depth, turbidity and tidal conditions, we decided to utilise two disparate types of fish visual system with known spectral sensitivities that are known to have chameleon prawns (or similar prawn species) in their diet[62,63]. Fish exhibit a range of variation in their visual capabilities[64], and so testing prawn camouflage using both dichromatic (i.e. possessing cones sensitive to short and medium wavelengths) and trichromatic (i.e. possessing cones sensitive to short, medium and long wavelengths) systems is important to account for camouflage perception to different receivers. Therefore, we map the digital images of prawns and seaweeds to corresponding models of predator vision[47,48], using the spectral sensitivity data found in the literature of two potential prawn predators: the dichromatic pollack (Pollachius pollachius), which has spectral peaks for single cones at 436 nm (short-wave sensitivity—SWS) and for double cones (paired cells with a similar morphology) at 521 nm (medium-wave sensitivity—MWS)[45], and the trichromatic two-spotted goby (Gobiusculus flavescens), which has spectral peaks for single cones at 456 nm (SWS) and for double cones at 531 nm (MWS) and 553 nm (long-wave sensitivity—LWS)[46]. Here, we propose that colour vision in pollack and goby is encoded by both single and double cones, but for the latter, we assumed that each double cone component works independently, as has been already reported for other fish species[65]. In addition, we incorporated 50% light transmission cut-off at 410 nm for both species[45,46] and used a D65 standard irradiance spectrum as a measure of incident illumination. Although this is not the precise light conditions under which natural predation would occur, water clarity is normally high in shallow tide pools (maximum 1.5 m deep) where prawns and seaweed were sampled.

We used a polynomial mapping function to convert prawn and seaweed images from the camera colour space to fish vision[47,48], generating values of pollack and goby cone catches. This is a widely used procedure for visual modelling and results in cone catch values that are in close accordance with data derived from spectrometry approaches[48,66,67]. We have previously characterized the spectral sensitivity of our camera in combination with the lens and filters[48,67]. Visual modelling resulted in multispectral images, which were used to estimate photon catch values for each colour channel in the regions of interest we selected. For the initial camouflage analysis, prawn's regions of interest consisted of the prawn carapace and abdomen from the area behind the eyes to the end of the

third abdominal somite, avoiding the stomach area. For the colour change analysis, prawn's regions of interest consisted of the entire prawn carapace and abdomen from the area behind the eyes to the end of the sixth abdominal somite where it reaches the telson. Regions of interest were also selected on images of green sea lettuce and red dulse, consisting of the entire fronds of small seaweed pieces.

**Quantification of prawn camouflage in the field**. In order to first predict whether the existing colour differences between wild prawns and seaweeds are perceived by fish predators, we used the widely implemented noise model for colour discrimination of Vorobyev and Osorio[49], which predicts chromatic signal discriminability as units of JNDs. Weber fractions were calculated based on the specific cone ratios of each visual model (shortest to longest wavelength; pollack 0.70: 1[45]; goby 0.72: 1: 0.60[46]) and a noise-to-signal ratio of 0.05 was used for the most abundant cone type in each species[68]. JNDs lower than 1 indicate that two colours cannot be discriminated by the receiver, while values higher than 1 would indicate a higher probability of prey detection by predators[49]. Using this model, we calculated colour differences (expressed as JNDs) between freshly sampled prawn types ($n = 64$ for both green and red colour types) and a single randomly selected image from a pool of 64 photos of both the green sea lettuce and red dulse, resulting in 64 JND values for each comparison. We predict that each prawn type would exhibit lower colour contrasts and be better camouflaged against the seaweed they are found mostly on in nature, that is, green chameleon prawns will be better concealed to green sea lettuce, while red individuals to the red dulse.

**Colour change trials**. This experiment was conducted to test the ability of chameleon prawns to change colour to improve their level of camouflage in response to non-matching algal substrates. Prawns ($n = 25$ of each green and red colour types) of similar size (here defined as the distance between the region behind the eyes and the junction between the sixth abdominal somite and the telson— green = 7.67 mm ± 0.34 [mean ± SE], red = 8.34 mm ± 0.41, $t_{(48)} = 1.28$, $p = 0.21$) were photographed in the field (day 0) and then were acclimatised to laboratory conditions for 24 h on matching algal substrates (green prawns on green sea lettuce, red prawns on red dulse). Prawns were then switched to algal substrates of contrasting coloration (green on red dulse, red on green sea lettuce). Photographs were then taken on days 5, 10, 15, 20, 25 and 30 to analyse the colour change process. During the first 10 days, when colour change was most abrupt (see Fig. 3), prawn survival was 100% for both colour types. Ninety two per cent of green and 88% of red prawns survived to day 20, after which we experienced an increased die off leaving us with 56% of green and 60% red prawns surviving to the end of the experiment. Missing values (i.e. data for prawns that died before the end of the experiment) were controlled using the option na.omit of the lmer function (see below), which allowed the inclusion of data from dead prawns in the model, but keeping only those from the time periods they were alive.

In order to evaluate the capacity of colour change in prawns along the experiment, we calculated a colour metric (hue) using two different approaches based on each fish visual system. For pollack, hue was simply defined as the SWS/ LWS ratio, since only these two colour channels are responsible for colour vision in this species[45]. For goby vision, we conducted a principal component analysis (PCA) to determine the main axis of colour variation that exists in prawn types and used this to determine a logical colour channel[14]. PCA was applied on a covariance matrix of the standardised cone catch data from the three reflectance channels and hue was further defined as the ratio [SWS/(MWS + LWS)], which is a simple measurement of short-wavelength versus medium–long-wave reflectance, broadly analogous to an opponent colour channel[14]. Similarly, in order to quantify how this change in colour relates to changes in camouflage, we used the same JNDs[49] procedure from the perspective of pollack[45] and goby[46] vision. To calculate these colour differences, each prawn image, for each day, was randomly paired with an image from a pool of 26 images of both green sea lettuce and red dulse (green prawns vs. red dulse, red prawns vs. green sea lettuce), allowing for comparison between prawn and seaweed regions of interest over the experiment.

**Behavioural choice trials**. Behavioural observations were concentrated around low-tide periods under a range of lighting conditions at Gyllyngvase Beach using custom-made Y-choice decision chamber to test whether green and red prawns actively choose an algal substrate that improved camouflage. Each individual was first acclimatised to the chamber for 1 min before the opaque divider was removed allowing the prawn to access the two arms of the y-choice chamber. For each trial lasting a maximum of 10 min, we recorded if the prawn individual made a choice and what substrate was chosen. Prawn decision within the allotted time was observed in 79 of 180 trials and did not differ between the two colour types ($X^2 = 0.22$, df = 1, $p = 0.64$).

**Statistics and reproducibility**. All statistical analysis were undertaken using the software R [v. 3.3.0][69] (the supporting data are available online as Supplementary Data 1 and the R scripts used for the statistical analysis can be assessed in https://github.com/rafaduarte87/chameleon-prawn-camouflage). Regardless of the visual model, a linear mixed-effects model was performed on colour JNDs for wild

prawns, with both prawn types (green or red) and seaweed species (green sea lettuce or red dulse) as fixed between-subjects factors, and 'individual' as a random factor to control for repeated measurements on the same individual, since each prawn was compared to both seaweed species. Similarly, for the colour change experiment, prawn hue and prawn JND were analysed separately for each colour type and visual model using a linear mixed-effect model, in which day was included as a fixed factor with seven levels (0, 5, 10, 15, 20, 25 and 30 days), size as a co-variate in order to control for possible colour changes due to prawn size, and individual as a random factor to account for repeated measurements in the same prawn individual across days. All linear models were fitted using the lmer function in the lme4 package[70]. Model residuals were checked for the homogeneity of variances and normal error distribution, for which colour JND data for green prawns in the colour change experiment needed a log transformation to meet model assumptions. In the case of significant effects, the Tukey's post hoc test was applied to compare mean differences between factor levels using the lsmeans function from the lsmeans package[70]. Finally, for the behavioural choice trial, the decisions of green and red prawns were analysed separately using an exact binomial test (binom.test function from the stats package in R), specifying the number of successful choices (green—green sea lettuce; red—red dulse) and the total sample size.

**Reporting summary**. Further information on research design is available in the Nature Research Reporting Summary linked to this article.

## Data availability

All data generated or analysed during this study are available in Supplementary Data 1.

## Code availability

The R scripts for statistical analysis can be assessed in https://github.com/rafaduarte87/chameleon-prawn-camouflage.

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

## Acknowledgements

Part of this study was funded by the Fundação de Amparo à Pesquisa do Estado de São Paulo (FAPESP), which granted a visiting researcher fellowship to R.C.D. during his Ph.D. (#2015/04484-8). We are grateful to Luis Robledo for his help on prawn sampling during the first part of this study and to Cyril Bennett for providing macro images of adult green and red prawns. We also thank Sara Mynott and Jim Galloway for providing additional images of prawns and seaweeds for the initial camouflage analysis.

## Author contributions

S.D.G., R.C.D. and M.S. designed the experiment. R.C.D. collected initial camouflage data. S.D.G., E.K. and N.A. collected behavioural data, S.D.G. collected colour change

data, R.C.D. and S.D.G. performed all image and statistical analysis. S.D.G., R.C.D. and M.S. contributed to the writing of the manuscript.

## Additional information

**Competing interests:** The authors declare no competing interests.

