## [Peer Review File · Communications Biology]

Reviewers' comments:

Reviewer #1 (Remarks to the Author):

This study shows how colour change and behaviour combine to improve camouflage in a marine prawn that shows both colour polymorphism and colour change. The study has three components: quantifying background match to different predator visual systems, testing the ability of prawns to change colour against different backgrounds, and testing behavioural preference for backgrounds that match their body colour. Many studies have looked at each of these components separately in a wide range of species but very few (if any) in combination. The combination of these components is powerful because it shows how individuals can use both colour change and behaviour to adjust their camouflage in response to varying environmental conditions. The main conclusions in this regard are supported by the data and will be of interest to those working in the field of animal coloration and camouflage. There is now good evidence showing how species with fixed coloration use behaviour to enhance camouflage and equally there is good evidence showing how colour change enhances camouflage against different backgrounds. The primary novelty of the study lies in showing how a colour changing species can use both colour change and behaviour to improve camouflage in complex and changing environments.

I have a couple of suggestions about how the manuscript can be improved.

1. In general, the introduction sets up the context and significance of the study well but is currently too long. It could easily be shortened from 6 to 4 paragraphs with little loss of information. The first paragraph is mostly unnecessary and there are excessive statements about limitations of previous work or lack of studies, some of which are debatable or unnecessary (e.g. lines 91 – 94). For example, we know that ontogenetic colour change for camouflage is very common (e.g. seasonal changes in plumage or pelage, metamorphosis between life stages etc), and this has been extensively documented in a broad range of taxa (cf lines 41 – 47). There are also many studies over many decades testing background choice to improve camouflage in a range of invertebrates, frogs, lizards (cf lines 49 – 50), though it is true that earlier studies had limitations in terms of quantifying camouflage or relationship to natural backgrounds (methods have advanced). In sum, I think the significance and 'knowledge gaps' can be stated more succinctly and positively. I also wonder if the hypotheses at the end of the introduction (lines 102 – 107) are necessary since they are so obvious, given the questions and experiments.

2. The methods are all sound except the images used for seaweed background were 'stock images' and presumably unstandardized. As I understand it, the images of the prawns were carefully calibrated to reflectance, but not the images of the background? Since background matching is estimated as the contrast between prawn and background, the use of uncalibrated background images will affect JND estimates, so precise interpretations of the estimated JNDs (e.g. below 1 being indistinguishable etc) are questionable. I think the use of stock images needs to be justified and if there was no alternative, then the limitations that this introduces need to be acknowledged.

3. The discussion was great, especially discussion of results in the context of the biology/natural history of the prawns and possible mechanisms. I did wonder about the polymorphism in the species – it suggests that background matching is likely to be only one factor in predator defence and that other factors are likely at play – e.g. frequency dependent selection from predator search image formation.

Reviewer #2 (Remarks to the Author):

Brief Summary:

This manuscript presents an examination of camouflage strategies in the chameleon prawns (*Hippolyte varians*). First the authors quantify the degree of colour matching for green and red colour morphs against natural green (sea lettuce) and red (dulse) backgrounds. Next the authors test whether the prawns can change colour to match their background by placing them on the non-matching background and photographing them at 5 day intervals up to 30 days. Finally, a Y-choice decision chamber was used to examine whether prawns preferentially selected matching backgrounds over non-matching backgrounds (i.e., to improve crypsis). They find that red and green prawns do closely match red and green background, respectively, when considering predator vision models calibrated for both dichromatic pollack (*Pollachius pollachius*) and trichromatic two-spotted goby (*Gobiusculus flavescens*). Prawns also changed colour in response to the new (mismatched) substrate primarily through changes in the relative proportion of reflectance in the short-wave channel. Colour change was initially rapid (up to 10 days), after which the rate of change became non-significant (10-30 days). Finally, the behaviour test showed that prawns exhibited a significant preference for backgrounds that matched their current colour.

Overall Impression:

The manuscript is clear and well written. Hypotheses are clear and explicit. Experiments are designed well. Methods and statistics are contemporary and appropriate. No new experiments are needed to support their conclusions, however there is plenty of room for follow up research. The central questions addressed in the manuscript are interesting and valuable beyond their study system.

The authors review a number of useful examples where animals employ colour change or behaviour to increase background matching, but they miss some important literature on species that combine these two mechanisms for enhanced crypsis (see Specific Comment #1). The results presented in this manuscript are highly comparable to what has been seen in previous work with salamander larvae and fish (e.g., Garcia and Sih 2003 *Oecologia* 137: 131-139; Rodgers et al 2013 *J Exp Biol* doi: 10.1242/jeb.080879). Yet, in the absence of reference to this previous work the reader is left with the impression that the current work is addressing questions that have yet to receive any substantial empirical investigation. Perhaps more importantly, the fact that the results from the current manuscript are broadly similar to what have been observed in phylogenetically and ecologically distinct systems surely tells us something important about the evolution of these strategies. In my view the manuscript would benefit from some discussion of this work, how it compares to their own results, and what that might mean more broadly.

Specific Comments:

1. The authors may wish to look at colour change literature in salamander larvae, especially Garcia and Sih 2003 *Oecologia* 137: 131-139, as it is highly analogous to what has been presented here but not cited (see also Garcia et al 2003 *Can J Zool* 8: 710-715; Garcia et al 2004 *Ecol Appl* 14: 1055-1064). In fact, several of the broad conclusions from the current manuscript were demonstrated by Garcia and Sih (2003). Specifically, they showed that 1) when there is no variation in background color available mismatched salamander larvae exhibited color change to better match the available background, and 2) when given a choice of background the larvae selected the colour-matched background. Moreover, Garcia and Sih (2003) also examined behavioural compensation for colour mismatch via refuge use, and whether the rate of colour change is influenced by perceived predation risk. On lines 234-237, the authors identify areas for future work that have already received some

investigation (e.g., by Garcia and Sih 2003). Notably, the current MS employs the substantially more rigorous contemporary methods for quantifying colour matching (e.g., with explicit and quantitative consideration of ecologically relevant predator vision), whereas earlier work with salamander larvae did not. In my view the manuscript would benefit from the authors revisiting some of the text in the introduction (e.g., lines 51-54, 62-64), discussion (e.g., lines 234-237), and should compare their results to this previous body of work in their discussion once they have read these articles.

2. Figures: While the colour palate for the figures is intuitive given the experimental design, it is likely to be problematic for red-green colour-blind readership, or when viewed in greyscale. If the authors want to retain the red-green colours, perhaps they could consider using a textured fill for one of the treatments in Figures 2 and 5. One of the lines in Figure 3 could be easily changed to a dashed line as well.

3. Lines 91-94: Suggest rewording. Strictly speaking, this MS does not explicitly examine how colour change and behavioural choices interact (although the authors identify a future research direction on lines 234-237 where this would be the case). Really they are looking at complimentary strategies that work of different time scales. The authors accurately describe in the last sentence of the Abstract.

Reviewer #3 (Remarks to the Author):

Title: first phrase is a bit sophomoric ... first-person verb tense is odd too. Suggest perhaps make the title more explicit and informative

Abstract:

Line 14 ... delete "dramatically" because it is so slow and that is not very dramatic.

Intro:

Line 40 for "rapid" pattern change you need to broaden the scope of fishes beyond the two that your lab has done (ref 30, 31). Allen et al 2015 showed truly rapid change in marine filefish and Tyrie et al 2015 also showed very rapid change in flounders. There are numerous refs in those two papers about other fishes that change within a few seconds.

Allen, J. J., Akkaynak, D., Sugden, A. & Hanlon, R. T. (2015). Adaptive body patterning, 3D skin morphology and camouflage measures of the slender filefish *Monocanthus tuckeri* on a Caribbean coral reef. *Biological Journal of the Linnean Society*, 116, 377-396.

Tyrie, E. K., Hanlon, R. T., Siemann, L. A. & Uyarra, M. C. (2015). Coral reef flounders, *Bothus lunatus*, choose substrates on which they can achieve camouflage with their limited body pattern repertoire. *Biological Journal of the Linnean Society*, 114, 629-638.

Line 51: after "morphological" add "and physiological"

Lines 65- 67 ... as an example you could reference cephalopods (ref 65 Hanlon et al 2009) but also use that ref with respect to your phrase "rapidly alter appearance regardless of the background." You could reference Allen et al 2010 showing experimentally that cuttlefish do not exhibit any substrate preference to camouflage on.

Allen, J. J., Mähger, L. M., Barbosa, A., Buresch, K. C., Sogin, E., Schwartz, J., Chubb, C. & Hanlon,

R. T. (2010). Cuttlefish dynamic camouflage: responses to substrate choice and integration of multiple visual cues. *Proceedings of the Royal Society B-Biological Sciences*, 277, 1031-1039.

Line 76: In addition to the lab experiments you cite for flatfish, you could add field study on marine flatfish that rapidly chose only the backgrounds that they could match. Tyrie et al 2015. This reference could complement that of Ref 44 on Line 79

Line 97: delete the term "Here"

Results:

These seem straightforward and compelling. However, I am not savvy with statistics so cannot comment of their detailed validity.

I like all the figures.

Discussion:

Line 94: good point about waves dislodging prawns

Line 203 ... the term "red dulse" is useful ... earlier in the ms the term dulse is used alone and few readers outside of England will know that this is red in color. Join the terms each time used in the ms for clarity.

Overall the Discussion is excellent and informative. There is a tad of redundancy in the beginning of the discussion but tolerable.

One minor omission to consider: is there anything at all known about the pigments involved in prawn color change? Worth mentioning for those interested in skin colour change. What about hormones?

Methods:

The color correction and interpretation in visual capabilities of fish are published and used by the Stevens group and seem to be robust and appropriate given the paucity of data available on predator color vision. It is quite useful for the wider community to have these two fish species added to the small repertoire of marine fishes with known color.

Line 419. What were these natural lighting conditions ... was it sunny, cloudy Late day or midday, etc.?

Line 423-425. Why did you limit the choice trial to 10 min? Fewer than half of the prawns decided in that short period. Please explain

Overall summary:

A very nice and useful study. There is novelty in the approach and stimulating discussion about larger questions to be addressed for camouflage tactics in general. Congrats. The community will appreciate the combo of field and lab studies.

Roger Hanlon

Reviewer #4 (Remarks to the Author):

The field of camouflage has long overlook the combined effect of colour change and behavioural choice on concealment. This paper has a perfect model system to experimentally test these ideas. It is a well written paper, with clear figures. I suggest this paper be accepted with minor revisions.

Authors quantify colour matching based on a visual model of proxy predators (e.g., pollack and gobi). An understandable limitation of the paper is that these proxy predators might have different colour vision from the shrimp's ecologically relevant predators. Authors should emphases how much colour vision varies between these vision models and ecologically relevant predators? Please make this limitation more explicit in the paper.

Without data and visual modeling of the spatial acuity of the predators (i.e., the modulation transfer function [MTF]), it is problematic to completely assess overall camouflage matching, as perception of pattern/texture is a missing dimension. This is a universal problem in the field of camouflage which has a focus on colour over texture. Authors should mention the limitation of ignoring the effect of the shrimp's colour changes on texture matching.

Would authors like to speculate in the discussion section why size affected the precision of colour change (e.g., smaller green shrimps colour change yields a better match to background than large green shrimps)? Do larger size shrimps have less selection acting on them, due to lower risk from predators because of more effective escape behaviours? Is there an ecological/seasonal reason why green shrimps achieve worse concealment when adjusting to the red background? Does this change in the natural habitat occur over a longer period of time? Does it occur when there are less predators? Alternatively, depending on the effect size, this might just be a statistical artifact.

It would be beneficial for the reader to be provided with some additional natural history details. Does the seasonal change in substrate correspond with a change in the behavior of shrimps and their exposure to different predator species.

In the methods section, state the R package and functions that were used to run these statistical tests. Further, for reproducibility of the statistical modeling, please provide a supplementary file for the GLM and LLM models fitted in R.

Dealing with missing data can introduce a source of bias. Please clarify the sentence: "Missing values were controlled for in statistical analyses."

Line 228: as the citations in text are superscripts, I recommend replacing 1m^2 with $1\text{m}^2\text{-squared}$

Remove the following sentence: "As natural habitats and environmental conditions are subject to increasing levels of human disturbance and climate change, understanding how the plasticity of animal coloration and the associated behaviours may enable species to cope with such changes is a key focus for future studies of evolutionary ecology." This feels like over selling the promise of research on camouflage for conservation science. Given the climate crises that our world faces, my opinion is researching camouflage with the pretense for addressing climate change is a form of green washing. I would like to thank the authors for an excellent piece of science, and I look forward to seeing this experimental paradigm be further developed in future studies.

Reply to Referees

We are very grateful to the four referees for their helpful and constructive comments and are delighted that they are enthusiastic about the work. We have addressed each comment below (**bold font**) and believe the manuscript is much improved as a result. We have also reduced the length of the introduction and abstract, and cut the number of references to 70 in order to comply with journal guidelines. Line numbers refer to the revised manuscript (clean version, without tracking changes).

Reviewer #1

Remarks to the Author:

This study shows how colour change and behaviour combine to improve camouflage in a marine prawn that shows both colour polymorphism and colour change. The study has three components: quantifying background match to different predator visual systems, testing the ability of prawns to change colour against different backgrounds, and testing behavioural preference for backgrounds that match their body colour. Many studies have looked at each of these components separately in a wide range of species but very few (if any) in combination. The combination of these components is powerful because it shows how individuals can use both colour change and behaviour to adjust their camouflage in response to varying environmental conditions. The main conclusions in this regard are supported by the data and will be of interest to those working in the field of animal coloration and camouflage. There is now good evidence showing how species with fixed coloration use behaviour to enhance camouflage and equally there is good evidence showing how colour change enhances camouflage against different backgrounds. The primary novelty of the study lies in showing how a colour changing species can use both colour change and behaviour to improve camouflage in complex and changing environments.

Thank you for your positive comments. We have revised the manuscript to address all of your concerns and are confident it is much improved.

I have a couple of suggestions about how the manuscript can be improved.

1. In general, the introduction sets up the context and significance of the study well but is currently too long. It could easily be shortened from 6 to 4 paragraphs with little loss of information. The first paragraph is mostly unnecessary and there are excessive statements about limitations of previous work or lack of studies, some of which are debatable or unnecessary (e.g. lines 91 – 94). For example, we know that ontogenetic colour change for camouflage is very common (e.g. seasonal changes in plumage or pelage, metamorphosis between life stages etc), and this has been extensively documented in a broad range of taxa (cf lines 41 – 47). There are also many studies over many decades testing background choice to improve camouflage in a range of invertebrates, frogs, lizards (cf lines 49 – 50), though it is true that earlier studies had limitations in terms of quantifying camouflage or relationship to natural backgrounds (methods have advanced). In sum, I think the significance and 'knowledge gaps' can be stated more succinctly and positively. I also wonder if the

hypotheses at the end of the introduction (lines 102 – 107) are necessary since they are so obvious, given the questions and experiments.

In order to reduce Introduction, we have removed various sentences and rewritten parts, including combining some paragraphs. This was also needed to comply with the journal guidelines that the Introduction should be less than 1000 words. Otherwise, we think that parts of the first paragraph are important because it provides to the reader general information about coloration and crypsis, appointing different solutions that many animals use to survive in colour heterogeneous backgrounds (including colour change and behavioural choices that are covered in this MS). We have nonetheless condensed this. We also preferred to keep our hypothesis at the end of this section because although obvious they link our questions with the Results (and in Communications Biology papers, the Results section appears before the Methods) but have again made these shorter / more direct. Finally, based in the comments addressed by the other referees (see below) we have added some new examples regarding certain theoretical points that required clarification.

2. The methods are all sound except the images used for seaweed background were ‘stock images’ and presumably unstandardized. As I understand it, the images of the prawns were carefully calibrated to reflectance, but not the images of the background? Since background matching is estimated as the contrast between prawn and background, the use of uncalibrated background images will affect JND estimates, so precise interpretations of the estimated JNDs (e.g. below 1 being indistinguishable etc) are questionable. I think the use of stock images needs to be justified and if there was no alternative, then the limitations that this introduces need to be acknowledged.

Thank you for highlighting this point of confusion. The seaweed images used in our analysis were not stock ones but ones that we acquired in the same manner as those of the prawns, as described in the ‘Photography’ section. As such both the prawns and background images were fully calibrated and standardised in the same way. We have removed the word ‘stock’ to avoid confusion (line 379).

3. The discussion was great, especially discussion of results in the context of the biology/natural history of the prawns and possible mechanisms. I did wonder about the polymorphism in the species – it suggests that background matching is likely to be only one factor in predator defence and that other factors are likely at play – e.g. frequency dependent selection from predator search image formation.

Thanks for this comment and we agree that frequency-dependent processes may be important too. We suggest that this species is polyphenic since the different phenotypes are plastic in terms of coloration and not fixed (i.e. they are not genetically determined true ‘morphs’ in the strict sense), and that both spatial and temporal heterogeneity of backgrounds are the likely driving forces maintaining the different colour forms in prawn population through differential

predation by fish. However, this does not exclude additional processes (e.g. predator search images), and we have added a short statement to this effect in the Discussion.

“In addition, we might also expect other processes to come into play for maintaining colour variation, including multiple morph types acting to hinder predator search image formation, and frequency-dependent selection⁵⁵”. (lines 235-237)

55. Bond, A. B. & Kamil, A. C. Visual predators select for crypticity and polymorphism in virtual prey. Nature 415, 609–613 (2002).

Reviewer #2

Remarks to the Author:

This manuscript presents an examination of camouflage strategies in the chameleon prawns (*Hippolyte varians*). First the authors quantify the degree of colour matching for green and red colour morphs against natural green (sea lettuce) and red (dulse) backgrounds. Next the authors test whether the prawns can change colour to match their background by placing them on the non-matching background and photographing them at 5 day intervals up to 30 days. Finally, a Y-choice decision chamber was used to examine whether prawns preferentially selected matching backgrounds over non-matching backgrounds (i.e., to improve crypsis). They find that red and green prawns do closely match red and green background, respectively, when considering predator vision models calibrated for both dichromatic pollack (*Pollachius pollachius*) and trichromatic two-spotted goby (*Gobiusculus flavescens*). Prawns also changed colour in response to the new (mismatched) substrate primarily through changes in the relative proportion of reflectance in the short-wave channel. Colour change was initially rapid (up to 10 days), after which the rate of change became non-significant (10-30 days). Finally, the behaviour test showed that prawns exhibited a significant preference for backgrounds that matched their current colour.

Overall Impression:

4. The manuscript is clear and well written. Hypotheses are clear and explicit. Experiments are designed well. Methods and statistics are contemporary and appropriate. No new experiments are needed to support their conclusions, however there is plenty of room for follow up research. The central questions addressed in the manuscript are interesting and valuable beyond their study system.

Thank you for your positive feedback.

5. The authors review a number of useful examples where animals employ colour change or behaviour to increase background matching, but they miss some important literature on species that combine these two mechanisms for enhanced crypsis (see Specific Comment #1). The results presented in this manuscript are highly comparable to what has been seen in previous work with salamander larvae and fish (e.g., Garcia and Sih 2003 *Oecologia* 137: 131-139; Rodgers et al 2013 *J Exp Biol* doi: 10.1242/jeb.080879). Yet, in the absence of reference to this previous work the reader is

left with the impression that the current work is addressing questions that have yet to receive any substantial empirical investigation.

Thank you for these suggestions – we agree that this work is valuable and relevant. Our specific reply for this comment is given below (comments #6 and #7).

6. Perhaps more importantly, the fact that the results from the current manuscript are broadly similar to what have been observed in phylogenetically and ecologically distinct systems surely tells us something important about the evolution of these strategies. In my view the manuscript would benefit from some discussion of this work, how it compares to their own results, and what that might mean more broadly.

This is a good point that certainly deserved more attention in the manuscript. We added a new sentence in the end of the Discussion framing our results in a more comparative view:

“The growing number of studies testing how combinations of chromatic (particularly colour change) and behavioural traits influence crypsis, and the fact that the above-mentioned traits are displayed by a range of phylogenetically and ecologically distinct systems, is indicative of the convergent evolution of these cryptic stratagems and the importance of adaptive benefits conveyed to species in order to maintain crypsis in heterogeneous habitats in wild systems”
(lines 289-293)

7. The authors may wish to look at colour change literature in salamander larvae, especially Garcia and Sih 2003 *Oecologia* 137: 131-139, as it is highly analogous to what has been presented here but not cited (see also Garcia et al 2003 *Can J Zool* 8: 710-715; Garcia et al 2004 *Ecol Appl* 14: 1055-1064). In fact, several of the broad conclusions from the current manuscript were demonstrated by Garcia and Sih (2003). Specifically, they showed that 1) when there is no variation in background color available mismatched salamander larvae exhibited color change to better match the available background, and 2) when given a choice of background the larvae selected the colour-matched background. Moreover, Garcia and Sih (2003) also examined behavioural compensation for colour mismatch via refuge use, and whether the rate of colour change is influenced by perceived predation risk. On lines 234-237, the authors identify areas for future work that have already received some investigation (e.g., by Garcia and Sih 2003). Notably, the current MS employs the substantially more rigorous contemporary methods for quantifying colour matching (e.g., with explicit and quantitative consideration of ecologically relevant predator vision), whereas earlier work with salamander larvae did not. In my view the manuscript would benefit from the authors revisiting some of the text in the introduction (e.g., lines 51-54, 62-64), discussion (e.g., lines 234-237), and should compare their results to this previous body of work in their discussion once they have read these articles.

We agree with the referee that the work of Garcia and Sih (2003) is important and should be cited – thank you for these suggestions. While the authors did not quantify salamander coloration and camouflage in the view of relevant predators, their work demonstrates an

important point discussed in our manuscript: how colour change and behavioural choices combine for improving animal crypsis. We include a brief statement of this reference (now ref #34) in our Introduction because we had to cut some parts of this section as suggested by referee #1 and to comply with journal guidelines.

“Tadpoles of two different salamander species, for example, change colour over a few hours when placed in mismatched substrates but the capacity for choosing concealing backgrounds differed between species and was affected by predation risk³⁴”. (lines 54-56)

We return to this paper (Garcia and Sih, 2003) in the Discussion section too, with the addition of a new paragraph discussing some avenues of research and referring to another paper suggested by the referee (Rodgers, 2013 – now ref #56).

“Changes in behavioural preferences mediated by modifications of body coloration have also been demonstrated in guppies (Poecilia reticulata), in which individuals spent significantly more time in black and white habitat zones after being induced to change colour in corresponding black and white tanks⁵⁶. Future work should further consider coloration and camouflage with regards to predator vision and measured attack rates. Another research avenue is to understand how predator cues may affect colour change and cryptic behaviours. For example, in the presence of a perceived predation threat animals may improve their capacity to change colour and select concealing backgrounds, the latter approach especially in slow colour-change species. In salamander larvae, the addition of predator cues in experimental tanks increases larval preference for dark backgrounds followed by a corresponding change in individual coloration³⁴. However, in the absence of predator cues, larvae spend equal time in light and dark habitat zones, adopting a more intermediate colour form³⁴. In the case of chameleon prawns, we would expect that the addition of predator cues may speed up the colour change process and lead to an increase in the proportion of prawns making a choice for concealing substrates. (lines 238-251)

8. Figures: While the colour palate for the figures is intuitive given the experimental design, it is likely to be problematic for red-green colour-blind readership, or when viewed in greyscale. If the authors want to retain the red-green colours, perhaps they could consider using a textured fill for one of the treatments in Figures 2 and 5. One of the lines in Figure 3 could be easily changed to a dashed line as well.

We have changed the brightness of both green and red colour from Fig. 2, 3 and 5. Also, we have added dashed lines to the green boxplot and bar plot (Fig. 1 and 5, respectively) and changed the green line in Fig. 2 to a dashed pattern as suggested.

9. Lines 91-94: Suggest rewording. Strictly speaking, this MS does not explicitly examine how colour change and behavioural choices interact (although the authors identify a future research direction on lines 234-237 where this would be the case). Really they are looking at complimentary strategies that work of different time scales. The authors accurately describe in the last sentence of the Abstract.

We have deleted ‘choices interact’ and replaced with ‘traits may operate in tandem’ and added ‘over different spatial and temporal scales’.

***“However, there is little information on how colour change and behavioural traits may operate in tandem in order to improve camouflage and reduce detection by predators over different spatial and temporal scales”.* (lines 73-75)**

Reviewer #3

Remarks to the Author:

10. Title: first phrase is a bit sophomoric ... first-person verb tense is odd too. Suggest perhaps make the title more explicit and informative

We take the point that not everyone likes titles of this nature, and also to comply with journal guidelines needed to alter this. The new title is: “The combined role of colour change and behavioural choice for camouflage in a marine prawn”, which we believe is more informative.

11. Line 14 ... delete “dramatically” because it is so slow and that is not very dramatic.

We have rewritten the abstract in order to meet journal guidelines and this no longer applies.

12. Line 40 for “rapid” pattern change you need to broaden the scope of fishes beyond the two that your lab has done (ref 30, 31). Allen et al 2015 showed truly rapid change in marine filefish and Tyrie et al 2015 also showed very rapid change in flounders. There are numerous refs in those two papers about other fishes that change within a few seconds.

- Allen, J. J., Akkaynak, D., Sugden, A. & Hanlon, R. T. (2015). Adaptive body patterning, 3D skin morphology and camouflage measures of the slender filefish *Monocanthus tuckeri* on a Caribbean coral reef. *Biological Journal of the Linnean Society*, 116, 377-396.

- Tyrie, E. K., Hanlon, R. T., Siemann, L. A. & Uyarra, M. C. (2015). Coral reef flounders, *Bothus lunatus*, choose substrates on which they can achieve camouflage with their limited body pattern repertoire. *Biological Journal of the Linnean Society*, 114, 629-638.

Thank you for these good suggestions. We inserted these two references (now refs #19 and #21) and another experimental study showing colour and pattern change in plaice fish (now ref #20). (line 26)

20. Kelman, E. J., Tiptus, P. & Osorio, D. Juvenile plaice (*Pleuronectes platessa*) produce camouflage by flexibly combining two separate patterns. *J. Exp. Biol.* 209, 3288–3292 (2006).

13. Line 51: after “morphological” add “and physiological”

This paragraph was changed and this no longer applies.

14. Lines 65- 67 ... as an example you could reference cephalopods (ref 65 Hanlon et al 2009) but also use that ref with respect to your phrase “rapidly alter appearance regardless of the background.”

You could reference Allen et al 2010 showing experimentally that cuttlefish do not exhibit any substrate preference to camouflage on.

Allen, J. J., Mäthger, L. M., Barbosa, A., Buresch, K. C., Sogin, E., Schwartz, J., Chubb, C. & Hanlon, R. T. (2010). Cuttlefish dynamic camouflage: responses to substrate choice and integration of multiple visual cues. *Proceedings of the Royal Society B-Biological Sciences*, 277, 1031-1039.

We added the Hanlon et al. 2009 reference in line 48 (now ref #32) and included an additional sentence using the Allen et al 2010 paper (now ref #33) to exemplify the lack of substrate preference in fast colour-changers such as cuttlefish.

“For example, cuttlefish (*Sepia officinalis*) do not exhibit substrate preference for camouflage, but rely on visual environmental cues to adopt camouflage patterns³³”. (lines 48-49)

15. Line 76: In addition to the lab experiments you cite for flatfish, you could add field study on marine flatfish that rapidly chose only the backgrounds that they could match. Tyrie et al 2015. This reference could complement that of Ref 44 on Line 79

Thanks for this suggestion. We have cited the Tyrie et al. 2015 reference (now ref #19) in the end of this paragraph in order to complement the reference #44 (now ref #31) to show that behavioural choices for concealing backgrounds were observed in fish using experiments both in the laboratory and in the field.

“Similarly, behavioural choices for colour matching backgrounds potentially improving camouflage were also demonstrated in fish species, both in the laboratory³¹ and in the field¹⁹”. (lines 59-61)

16. Line 97: delete the term “Here”

Removed.

17. These seem straightforward and compelling. However, I am not savvy with statistics so cannot comment of their detailed validity.

Thanks. Following the recommendation of referee #4 we have now included all R codes for the statistical analysis as a supplementary file. This will be important for the readers understand what functions and packages we used to analyse our data.

18. I like all the figures.

Thanks. Following the recommendation of referee #1 we have changed the colour of bars and lines in our figures in order to avoid problems with colour-blind readers.

19. Line 94: good point about waves dislodging prawns

Thanks.

20. Line 203 ... the term "red dulse" is useful ... earlier in the ms the term dulse is used alone and few readers outside of England will know that this is red in color. Join the terms each time used in the ms for clarity.

This is a good point, thanks. We have added the colour each time either seaweed species are mentioned.

21. Overall the Discussion is excellent and informative. There is a tad of redundancy in the beginning of the discussion but tolerable.

Thanks.

22. One minor omission to consider: is there anything at all known about the pigments involved in prawn color change? Worth mentioning for those interested in skin colour change. What about hormones?

In Gamble and Keeble's 1900 paper they discuss the presence of three perceived pigments (blue, red and yellow) within the chromatophore cells. The pigments determining coloration of caridean prawns were extensively studied by Bauer's papers in 80's, and summarized in his book in 2004 (we cited both 1981 paper and 2004 book in the MS). Within the book (specifically pg 97) Bauer discusses how the poorly understood blue colour may be associated with a red pigment. Additionally, he discusses the considerable variation in colour patterns in *H. varians* due to the dispersion and densities of pigments within a green brown chromatophore containing two pigments a 'yellowish green' and a 'reddish brown'. We presume that the pigments and chromatosomes determining chameleon prawn's coloration are similar to those described by Bauer in *Heptacarpus pictus* and *H. paludicola* (1981). In his work, Bauer describe four chromatosome types in these two species: red-white, red, yellow and red-yellow, all containing carotenoid-based pigments and a crystal-like material (responsible for the white colour). Depending on the concentration and dispersion of each pigment within the chromatosome, prawn would adopt a specific appearance.

There is also considerable information about the physiological control of colour change in crustaceans (reviewed in Stevens 2016 and Duarte et al 2017, the latter cited in our MS). We know that chromatic change in crustaceans are guided by vision and mediated by different hormones (released or not by the eyestalks). However, there is a lack of studies integrating both visual information and hormones on controlling colour change in animals, especially crustaceans.

We have discussed briefly the importance of the different pigments and chromatophores in the colour change process in replying to a comment from the referee #4 (see below). However, although all these information (pigments and hormones) are valuable, we they are not directly important here and we are limited by word count restrictions. In this paper we aim to provide the ecological aspects of coloration and camouflage in chameleon prawns and so we have only briefly included a summary of the above.

Methods:

23. The color correction and interpretation in visual capabilities of fish are published and used by the Stevens group and seem to be robust and appropriate given the paucity of data available on predator color vision. It is quite useful for the wider community to have these two fish species added to the small repertoire of marine fishes with known color.

This is true. Our paper and others dealing with visual modelling are important sources to increase our knowledge about colour vision in different animal groups.

24. Line 419. What were these natural lighting conditions ... was it sunny, cloudy Late day or midday, etc.?

The weather conditions were very variable owing to Cornwall having highly changeable weather conditions even within each day. This should not add any bias or confounds to the work since the studies were done over a number of days and time periods and all trials conducted within each experimental block. We included additional information in the Methods as follow:

“Behavioural observations were concentrated around low tide periods under a range of lighting conditions at Gyllyngvase Beach using custom made Y-choice decision chamber to test whether green and red prawns actively choose an algal substrate that improved camouflage.” (lines 442-444)

25. Line 423-425. Why did you limit the choice trial to 10 min? Fewer than half of the prawns decided in that short period. Please explain

The trial period was set at 10 minutes to allow for ample trials to be conducted using the y-choice decision chamber during low tide periods. Previous behavioural works cited in this manuscript have also used a trial period of 10 minutes (ref #18 Smithers *et al*, 2018 and ref #56 Rogers *et al*, 2013 for example). In a biological context we have mentioned how the demonstrated behavioural choices may aid in the maintenance of camouflage when a prawn is dislodged from its substrate by wave action. In this instance we would expect prawns to remedy this situation as quickly as possible, however we acknowledge the referee’s point and perhaps if more time was allowed for trials we may have had a greater proportion of decisions although our results clearly demonstrate a behavioural preference.

26. Overall summary. A very nice and useful study. There is novelty in the approach and stimulating discussion about larger questions to be addressed for camouflage tactics in general. Congrats. The community will appreciate the combo of field and lab studies. Roger Hanlon.

Thank you for your thorough and constructive feedback.

Reviewer #4

Remarks to the Author:

27. The field of camouflage has long overlook the combined effect of colour change and behavioural choice on concealment. This paper has a perfect model system to experimentally test these ideas. It is a well written paper, with clear figures. I suggest this paper be accepted with minor revisions.

Thanks for your positive feedback. We have provided amendments based in all your comments and from the other referees and hope this is now a better version of the manuscript.

28. Authors quantify colour matching based on a visual model of proxy predators (e.g., pollack and gobi). An understandable limitation of the paper is that these proxy predators might have different colour vision from the shrimp's ecologically relevant predators. Authors should emphases how much colour vision varies between these vision models and ecologically relevant predators? Please make this limitation more explicit in the paper.

Thanks for this comment. We agree that we need to make this information more explicit in the paper. Based on this, we included in the Methods section a brief statement on the importance to model prawn camouflage under different visual systems.

“Fish exhibit a range of variation in their visual capabilities, and so testing prawn camouflage using both dichromatic (i.e. possessing cones sensitive to short and medium wavelengths) and trichromatic (i.e. possessing cones sensitive to short, medium and long wavelengths) is important to account for camouflage perception to different receivers.” (lines 352-356)

64. Marshall, N. & Vorobyev, M. The design of color signals and color vision in fishes. in *Sensory Processing of the Aquatic Environment* (eds. Collin, S. & Marshall, N.) 194–222 (Springer-Verlag, 2003).

29. Without data and visual modeling of the spatial acuity of the predators (i.e., the modulation transfer function [MTF]), it is problematic to completely assess overall camouflage matching, as perception of pattern/texture is a missing dimension. This is a universal problem in the field of camouflage which has a focus on colour over texture. Authors should mention the limitation of ignoring the effect of the shrimp's colour changes on texture matching.

Thanks for this comment. We have indicated this in the Discussion section where we discuss transparent prawns, since the types of homogeneous prawns we study here do not have particularly pronounced patterns:

“In our study, the visual models used are based on colour perception and the spectral sensitivities of ecologically relevant predators available in the literature^{45,46}. However, we do not model the spatial acuity of the predators, which is relevant to pattern matching and something that may be especially relevant to transparent prawn types with their intricate markings.” (lines 262-265)

30. Would authors like to speculate in the discussion section why size affected the precision of colour change (e.g., smaller green shrimps colour change yields a better match to background than large green shrimps)? Do larger size shrimps have less selection acting on them, due to lower risk from predators because of more effective escape behaviours? Is there an ecological/seasonal reason why green shrimps achieve worse concealment when adjusting to the red background? Does this change in the natural habitat occur over a longer period of time? Does it occur when there are less predators? Alternatively, depending on the effect size, this might just be a statistical artifact.

Thanks for this comment. We agree that this relationship between colour change and size should be discussed in the text. We added a new paragraph in the Discussion, as follow:

“Our results also indicate that the effectiveness of colour change for camouflage was higher for small green prawns compared to larger individuals. This relationship needs to be properly investigated in future studies but speculatively could indicate that larger green prawns have less selection acting on them due to more effective escape behaviours or by achieving a size-refuge from predators, or due to physiological limitations. Why this occurs only for green prawns is difficult to explain but may be related to the fact that red prawns when changing to green always exhibit lower JNDs compared to the opposite (Fig. 3). This seems to be a physiological constraint, since the red coloration is probably defined by the presence of red-yellow pigments within chromatophore cells, while the green tone is provided by the presence of only the yellow pigment (similar to that observed in the prawns *Heptacarpus pictus* and *H. paludicola*³⁷). Therefore, changing from red to green may be easier and faster than the opposite since both pigments (i.e. red and yellow) are already present within the colour cells of red prawns. On the other hand, green prawns changing to red would need to metabolize red pigments (probably by food ingestion⁴¹) which would take more time, especially for larger individuals, potentially explaining the higher JNDs during the colour change process and the size-effects we observed.” (lines 177-191)

31. It would be beneficial for the reader to be provided with some additional natural history details. Does the seasonal change in substrate correspond with a change in the behavior of shrimps and their exposure to different predator species.

This is a very good question but unfortunately we do not have this information at present. In our study site the candidate predators (e.g. rock pool fish) are around all year though other species that may come in shore (e.g. mackerel) are much more seasonal. However we have not tested changes in behaviour with season yet, and have no specific reason to expect changes in predator exposure. More work would be needed here. We have added a brief statement to the Discussion.

“In addition to seasonal changes in substrate, intertidal species may undergo seasonal shifts in predation pressures as fish species move inshore and as juveniles develop, and future work could quantify how the level of crypsis may vary with these predator shifts.” (lines 216-219)

32. In the methods section, state the R package and functions that were used to run these statistical tests. Further, for reproducibility of the statistical modeling, please provide a supplementary file for the GLM and LLM models fitted in R.

Thanks for this comment. All functions and packages used were included in the Methods section. We also provide a supplementary file with the R script containing all statistical analysis we performed along the manuscript.

33. Dealing with missing data can introduce a source of bias. Please clarify the sentence: "Missing values were controlled for in statistical analyses."

We clarified this sentence and hope now it is easier to follow.

"Missing values (i.e. data for prawns that died before the end of the experiment) were controlled using the option 'na.omit' of the 'lmer' function (see below), which allowed the inclusion of data from dead prawns in the model but keeping only those from the time periods they were alive". (lines 415-418)

34. Line 228: as the citations in text are superscripts, I recommend replacing m^2 with m^2 squared.

Agreed - edited (line 227).

35. Remove the following sentence: "As natural habitats and environmental conditions are subject to increasing levels of human disturbance and climate change, understanding how the plasticity of animal coloration and the associated behaviours may enable species to cope with such changes is a key focus for future studies of evolutionary ecology." This feels like over selling the promise of research on camouflage for conservation science. Given the climate crises that our world faces, my opinion is researching camouflage with the pretense for addressing climate change is a form of green washing.

The sentence was removed.

36. I would like to thank the authors for an excellent piece of science, and I look forward to seeing this experimental paradigm be further developed in future studies.

We are pleased that you enjoyed our work. Thank you for you positive and constructive comments.

REVIEWERS' COMMENTS:

Reviewer #1 (Remarks to the Author):

The authors received four constructive reviews, and in my view, they have done a thorough job and the addressing the reviewer comments. I think that the manuscript now presents a balanced and insightful view of the combined role of colour change and behaviour in facilitating camouflage in variable environments. I have no further suggestions and look forward to seeing the paper published.

Reviewer #2 (Remarks to the Author):

The manuscript is a compelling and tractable study that examines a question that is of interest to a broad audience. In the revised manuscript the Introduction and Discussion sections are significantly improved and now do a much better job of placing this work and its conclusions in the broader context of previous research. Statistical analyses and experiential techniques are appropriate. The provision of annotated R code combined with the level of detail in the methods make the experiment and analyses reproducible.

The authors have satisfactorily addressed all of the comments I had raised (Reviewer #2), and have done a commendable job of address the the comments from the 3 other reviewers as well.

Line 54: Generally speaking "tadpole" is reserved for the larvae of frogs and toads. For salamanders, herpetologists simply refer to them as "larvae" during this life stage. Swapping out "Tadpoles" for "The larvae" will fix this very minor point.

Beyond this I have no further concerns and I congratulate the authors on a well executed piece of research.

Sincerely,

Thomas Hossie

Reviewer #3 (Remarks to the Author):

I am happy with this revised manuscript. The authors have made many changes that improved the ms substantially. The authors made revisions that I had suggested, so I am pleased with that portion of the product. I read the other reviews as well as the Response Letter for all reviews and these seem reasonable and effective.

I do have two other niggling (but of some importance) issues:

Line 16 ... ADD "... reducing the chance of prey detection OR RECOGNITION by the visual system"
Reason: these two factors are key to camouflage ... no good reason for you to ignore recognition (i.e. disruptive coloration, masquerade, mimicry).

Line 21 " ... to adjust THE COLOUR ASPECT of their camouflage ..."
Reason: color is just one part of camouflage ... there is pattern, contrast, 3D texture, behavior etc.

This paper will be a welcome addition to the literature and helps guide some future research.

Reviewer #4 (Remarks to the Author):

Authors have successfully addressed my comments.

I suggest this manuscript be accepted.

REVIEWERS' COMMENTS:

Reviewer #1 (Remarks to the Author):

The authors received four constructive reviews, and in my view, they have done a thorough job and the addressing the reviewer comments. I think that the manuscript now presents a balanced and insightful view of the combined role of colour change and behaviour in facilitating camouflage in variable environments. I have no further suggestions and look forward to seeing the paper published.

We are glad the reviewer is satisfied with our responses.

Reviewer #2 (Remarks to the Author):

The manuscript is a compelling and tractable study that examines a question that is of interest to a broad audience. In the revised manuscript the Introduction and Discussion sections are significantly improved and now do a much better job of placing this work and its conclusions in the broader context of previous research. Statistical analyses and experiential techniques are appropriate. The provision of annotated R code combined with the level of detail in the methods make the experiment and analyses reproducible. The authors have satisfactorily addressed all of the comments I had raised (Reviewer #2), and have done a commendable job of address the comments from the 3 other reviewers as well.

Line 54: Generally speaking "tadpole" is reserved for the larvae of frogs and toads. For salamanders, herpetologists simply refer to them as "larvae" during this life stage. Swapping out "Tadpoles" for "The larvae" will fix this very minor point. Beyond this I have no further concerns and I congratulate the authors on a well-executed piece of research.

Sincerely,

Thomas Hossie

We have changed 'tadpoles' to 'the larvae' - line 54

Reviewer #3 (Remarks to the Author):

I am happy with this revised manuscript. The authors have made many changes that improved the ms substantially. The authors made revisions that I had suggested, so I am pleased with that portion of the product. I read the other reviews as well as the Response Letter for all reviews and these seem reasonable and effective. I do have two other niggling (but of some importance) issues:

Line 16 ... ADD "... reducing the chance of prey detection OR RECOGNITION by the visual system" Reason: these two factors are key to camouflage ... no good reason for you to ignore recognition (i.e. disruptive coloration, masquerade, mimicry).

Line 21 " ... to adjust THE COLOUR ASPECT of their camouflage ..." Reason: color is just one part of camouflage ... there is pattern, contrast, 3D texture, behavior etc.

This paper will be a welcome addition to the literature and helps guide some future research.

We have made the appropriate changes to lines 16 and 22.

Reviewer #4 (Remarks to the Author):

Authors have successfully addressed my comments.

I suggest this manuscript be accepted.

Again, we are glad the reviewer is satisfied with our responses.